# MOTION-GROUNDED VIDEO REASONING: UNDERSTANDING AND PERCEIVING MOTION AT PIXEL LEVEL

## ABSTRACT

In this paper, we introduce **Motion-Grounded Video Reasoning**, a new motion understanding task that requires generating **visual answers** (video segmentation masks) according to the input question, and hence needs implicit spatiotemporal reasoning and grounding. This task extends existing spatiotemporal grounding work focusing on explicit action/motion grounding, to a more general format by enabling implicit reasoning via questions. To facilitate the development of the new task, we collect a large-scale dataset called **GROUNDMORE**, which comprises 1,673 video clips, 243K object masks that are deliberately designed with 4 question types (*Causal, Sequential, Counterfactual, and Descriptive*) for benchmarking deep and comprehensive motion reasoning abilities. GROUNDMORE uniquely requires models to generate visual answers, providing a more concrete and visually interpretable response than plain texts. It evaluates models on both spatiotemporal grounding and reasoning, fostering to address complex challenges in motion-related video reasoning, temporal perception, and pixel-level understanding. Furthermore, we introduce a novel baseline model named **Mo**tion-Grounded Video **R**easoning **A**ssistant (**MORA**). MORA incorporates the multimodal reasoning ability from the Multimodal LLM, the pixel-level perception capability from the grounding model (SAM), and the temporal perception ability from a lightweight localization head. MORA achieves respectable performance on GROUNDMORE outperforming the best existing visual grounding baseline model by an average of 28.8% relatively. We hope this novel and challenging task will pave the way for future advancements in robust and general motion understanding via video reasoning segmentation.

## 1 INTRODUCTION

Understanding motions (Aggarwal & Cai, 1999; Corona et al., 2020; Zhou et al., 2012; Tevet et al., 2022) in dynamic video scenes has long been an important topic in the computer vision community. It plays a crucial role in many vital real-world applications, such as scene/video understanding (Saleemi et al., 2010; Sturgess et al., 2009; Mottaghi et al., 2016; Tsai et al., 2011; Fan et al., 2018), autonomous driving (Chen et al., 2015; Singh et al., 2022; Leon & Gavrilescu, 2019; Hu et al., 2023), and human-computer interaction (Aggarwal & Park, 2004; Wren & Pentland, 1999; Schmidt, 2000). Existing motion understanding tasks (e.g., action recognition (Soomro et al., 2012; Carreira & Zisserman, 2017), temporal action localization (Caba Heilbron et al., 2015; Jiang et al., 2014), spatiotemporal action/object detection (Gkioxari & Malik, 2015; Gu et al., 2018; Li et al., 2021; Vu et al., 2018; Jiang et al., 2020), video object segmentation (Xu et al., 2018; Seo et al., 2020; Khoreva et al., 2019; Cheng et al., 2023b; Ding et al., 2023)) are designed to either comprehend spatial interactions or detect motions in temporal span.

However, motion is a complex spatiotemporal concept involving interactions between visual entities over time. Understanding motion-related attributes abstracted from dynamic scenes is crucial for comprehensive motion understanding. Table 1 highlights that existing tasks only address this challenge from specific aspects. As shown in Figure 1(a), action recognition focuses on identifying actions within a curated video clip, primarily using spatial features. The models are not required to distinguish fine-grained motion patterns over time but to recognize "the motion" mostly based on spatial features in a temporal-agnostic (Huang et al., 2018) manner due to potential single-frame bias (Lei et al., 2022). It leads to overlook fine-grained temporal motion patterns. Conversely, temporal action localization in Figure 1(b) emphasizes the temporal dimension but lacks detailed

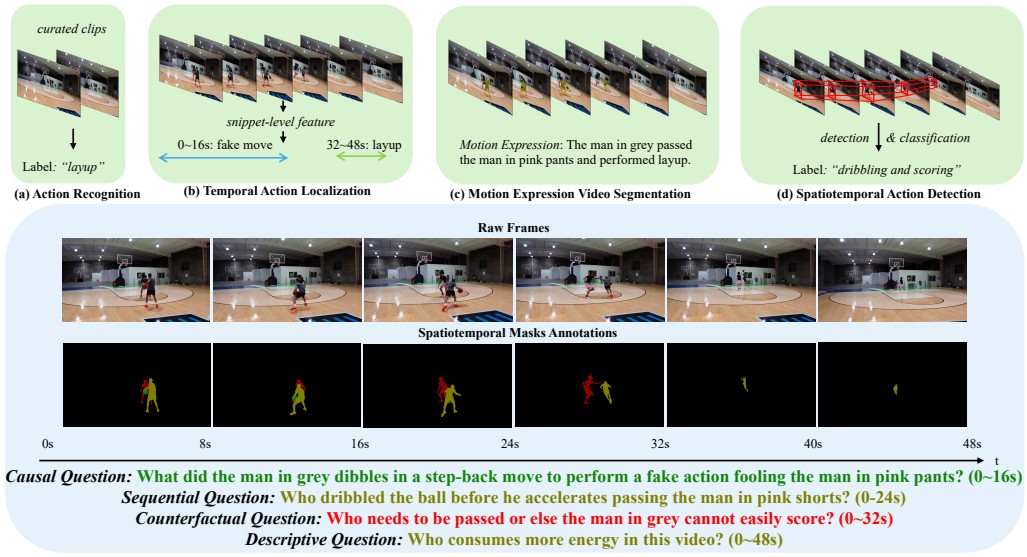

Figure 1: The illustration of the comparison between our **Motion-Grounded Video Reasoning** and previous video motion understanding tasks. Existing video motion understanding tasks (a)-(d) could at most address one or two key problems, either lacking fine-grained spatiotemporal perception or ignoring motion-related reasoning. (e) Our Motion-Grounded Video Reasoning considers both subject and object in motion as well as temporally adjacent events, performing challenging reasoning given four types of questions (***Causal, Sequential, Counterfactual, and Descriptive***) carefully designed in our GROUNDMORE dataset and output **spatiotemporal masks** to indicate the answer visually at the **pixel level**. For instance, in the question "who needs to be passed or else the man in grey cannot easily score?", the motion "pass" and the subject "the man in grey" as well as an adjacent event "easily score" are provided in this question, the model needs reason about the object "the man in pink shorts", while output spatiotemporal masks (only between 0 to 32s where the motion "pass" happens). Such a paradigm fully grasps the spatiotemporal contexts of motion and provides an explainable response to evaluate the motion understanding ability. The colors of the questions are corresponded to the spatiotemporal masks.

spatial analysis at the object level, relying on snippet-level features. Spatiotemporal action detection aims to localize actions in both dimensions but typically focuses only on humans in predefined actions (*e.g.*, AVA (Gu et al., 2018), MultiSports (Li et al., 2021)), neglecting other interacting objects. It impairs the integrity of the spatial perception of motion understanding. Previous compositional action recognition investigates subject-object interaction and examines whether the model could distinguish pretended actions, but the benchmark (Goyal et al., 2017) only contains short clips, making the task fall short in analyzing the temporal context of motions. Thus, a crucial question arises: *What will be a more comprehensive task for motion understanding?* Inspired by the recent reasoning segmentation task in image domain (Lai et al., 2023), and considering the spatiotemporal nature of the motion as mentioned above, a feasible answer is to design an implicit video reasoning segmentation task where all necessary spatial and temporal factors of the motion of interest are taken into account, and then the motion-related object, which could be viewed as the medium of the corresponding motion, will be masked out as the final response.

First, understanding specific motions requires analyzing their spatial contexts. For instance, in the interaction scenario "a boy kicked the ball for entertainment", the entities "a boy" and "the ball" constitute the spatial context for the motion "kicked". A comprehensive understanding of "kicked" involves grasping the interaction tuple <a boy, kick, the ball>. While spatiotemporal action localization tasks might address this problem, current benchmarks (*e.g.*, AVA (Gu et al., 2018)) focus primarily on human-centric cases and overlook the bidirectional nature of interactions. A more effective approach would involve a question-answering format that leverages motion-related objects to visualize and reason about the interaction, enhancing spatial understanding. Second, temporal context, which provides chronological order to distinguish

Table 1: Comparison of different motion understanding tasks. **Spatial Context** means whether to consider object-level interaction, **Temporal Context** indicates the influence of temporally adjacent motions/events, **Motion Abstraction** means understanding of motion-related abstract attributes, **Pixel-level Output** means whether output object segmentation mask as the final response and **Implicit Reasoning** means the ability to understand textual input without explicit object information.

| Tasks | Datasets & Benchmarks | Spatial Context | Temporal Context | Motion Abstraction | Pixel-level Output | Implicit Reasoning |
|---|---|---|---|---|---|---|
| Action Recognition | Kinetics400 (Carreira & Zisserman, 2017), UCF101 (Soomro et al., 2012) | ✗ | ✗ | ✗ | ✗ | ✗ |
| Temporal Action Localization | ActivityNet (Caba Heilbron et al., 2015), THUMOS14 (Jiang et al., 2014) | ✗ | ✓ | ✗ | ✗ | ✗ |
| Spatiotemporal Action Localization | AVA (Gu et al., 2018), MultiSports (Li et al., 2021) | ✓ | ✓ | ✗ | ✗ | ✗ |
| Motion Expression Video Reasoning | MeViS (Ding et al., 2023) | ✓ | ✗ | ✗ | ✓ | ✗ |
| Motion-Grounded Video Reasoning | GROUNDMORE (Ours) | ✓ | ✓ | ✓ | ✓ | ✓ |

different motions, is also crucial for motion understanding. Temporal information not only delineates temporal boundaries but also enables understanding of cause-and-effect relationships between actions. For example, in "`the woman opened the refrigerator before taking out the milk`", the two motions are connected, necessitating understanding of both for full comprehension. Thus, a question-answering paradigm can be designed, where a complete scene description with spatiotemporal context is converted into a motion-related question. However, merely answering the question cannot fully convey motion understanding, as language alone, if not visually grounded, is not the most direct explanation of visual concepts (Glenberg & Kaschak, 2002), and temporal information cannot be precisely represented by words (Xiao et al., 2024).

To address these issues and facilitate comprehensive motion understanding, we introduce a novel task: **Motion-Grounded Video Reasoning** as illustrated in Figure 1(e). This task requires models to take the motion-related question along with the video as input and output spatiotemporal segmentation masks of a specific object as a **pixel-level** visual answer. Such detailed spatiotemporal grounding allows for advanced motion comprehension. To further evaluate versatile spatiotemporal reasoning, we carefully design four types of questions in our newly collected dataset GROUNDMORE (**Ground**ing via **Mo**tion **Re**asoning). As shown in Figure 1(e), *Causal* questions explore the motivations behind motions, *Sequential* questions probe the order of temporally adjacent motions, *Counterfactual* questions are designed for imagining and reasoning about false reality and *Descriptive* questions ask about the general dynamic scene or abstract motion-related attributes such as *enregetic, naughty, excited, etc.* GROUNDMORE consists of about **1,673 video clips**, **7,301 questions** and **243K object masks** involving **3,942 different objects**, ensuring a robust evaluation of motion understanding. Additionally, our task aligns with Video Object Segmentation (VOS) (Xu et al., 2018; Ding et al., 2023) but introduces additional challenges: 1) the use of implicit question inputs versus explicit referring expressions, and 2) the requirement for spatiotemporal object masks rather than spatial-only (no temporal localization requirement in current RVOS datasets), emphasizing the need for accurate temporal perception. We emphasize the practical benefits of the new task in diverse real-world applications. For example, localizing potential threats in public transportation often involves ambiguous information about the suspects. A robust Motion-Grounded Video Reasoning system can address this by processing queries like "`Who is acting suspiciously in this airport?`", effectively identifying unusual behaviors with implicit reasoning and spatiotemporal grounding.

We conduct an extensive evaluation for various image/video grounding baselines on GROUNDMORE, though scoring competitive performances in other benchmarks (Kazemzadeh et al., 2014; Xu et al., 2018; Ding et al., 2023), none of them performs satisfyingly on our new task as shown in Table 3. Considering the spatiotemporal reasoning and grounding nature of the task, we further propose a new baseline model called **Mo**tion-Grounded Video **R**easoning **A**ssistant (**MORA**). MORA integrates LLaVA (Liu et al., 2023a), which is capable of complex multimodal reasoning, as the reasoning module, and a pretrained SAM (Kirillov et al., 2023) decoder as the mask head. To further empower the model of temporal awareness, we additionally introduce a novel **[LOC]** token for temporal information embedding and add a temporal localization head to decode a binary temporal mask; thus inhibiting false temporal activation during spatiotemporal mask decoding. Our MORA achieves overall SOTA performance on the proposed GROUNDMORE, but there still remains a large room for future improvement (e.g., HTR (Miao et al., 2024) could reach 67.1 with $\mathcal{J}\&\mathcal{F}$ metric on Ref-YouTubeVOS as its SoTA, while only 10.41 on GROUNDMORE), which also underscores the increased difficulty of GROUNDMORE.

Our contributions are as follows:

- We introduce a new task, ***Motion-Grounded Video Reasoning***, designed to assess multimodal models' reasoning and perception capabilities for motion understanding, filling the gap between referring VOS/action detection and motion-related video reasoning.
- We collect a large-scale and versatile video dataset, named **GROUNDMORE** for the proposed Motion-Grounded Video Reasoning task.
- We comprehensively evaluate existing image/video grounding baseline models on our GROUND-MORE, revealing their deficient motion understanding abilities. On the other hand, our proposed MORA method achieves **SOTA** performance on GROUNDMORE. The results also suggest substantial room for future improvement.

## 2   RELATED WORK

**Motion Understanding in Videos.** Motion understanding is pivotal in video analysis, serving as the basis for interpreting dynamic scenes and activities. Action recognition (Carreira & Zisserman, 2017; Soomro et al., 2012) identifies specific actions in videos, while temporal action localization (Caba Heilbron et al., 2015; Jiang et al., 2014) pinpoints the exact time intervals of these actions, requiring a thorough grasp of motion patterns over time. Spatiotemporal action detection (Gkioxari & Malik, 2015; Gu et al., 2018; Li et al., 2021) and video object detection (Vu et al., 2018; Jiang et al., 2020) predict object bounding boxes in both spatial and temporal domains. Video object segmentation (VOS) (Xu et al., 2018) and video tracking (Cheng et al., 2023b) capture moving objects in videos relying on objects appearance. To fully understand motion, it is crucial to comprehend its spatiotemporal contexts, including the involved objects and temporally adjacent information. In this paper, we introduce Motion-Grounded Video Reasoning, a new task that aims to reason based on the spatiotemporal context of motion and respond with video object masks.

**Spatiotemporal Video Grounding.** Spatiotemporal video grounding involves leveraging temporal cues to localize, identify, and interpret objects based on natural language expressions. Existing pipelines either focus on enhancing visual/textual semantic understanding (Baradel et al., 2018; He & Ding, 2024; Khoreva et al., 2019; Miao et al., 2024; Lin et al., 2023; Li et al., 2023) or strengthening cross-modal interaction (Wu et al., 2023; Gu et al., 2024; Ding et al., 2022; Liu et al., 2021; Wu et al., 2022a;b; Miao et al., 2023). Action grounding (Regneri et al., 2013; Zeng et al., 2020) localizes actions indicated by the input descriptions, and referring VOS (Seo et al., 2020; Khoreva et al., 2019) aims to ground objects at pixel level based on object-related expressions and recent work MeViS (Ding et al., 2023) introduces more challenging motion expressions, demanding advanced motion understanding to segment moving objects. These advanced frameworks achieve outstanding performance in grounding objects of interest in both spatial and temporal dimensions, however, these works primarily focus on context-level understanding and cannot perform complex reasoning and motion context perceiving. Recent works (Lai et al., 2023; Huang et al., 2024; Munasinghe et al., 2023; Zhang et al., 2024a; Rasheed et al., 2024; Zhang et al., 2024b) connects reasoning abilities of LLMs to the grounding task. PG-Video-LLaVA (Munasinghe et al., 2023) is a video-LLM equipped with pixel-level grounding modules but struggles with implicit reasoning/referring. LITA (Huang et al., 2024) leverages LLM for 1-D video temporal span localization with text query. In this paper, we present a novel baseline model, MORA, that handles both complex spatiotemporal reasoning and grounding for the proposed Motion-Grounded Video Reasoning task.

**Video Reasoning.** Video reasoning (Wang et al., 2024a; Wu et al., 2021; Tapaswi et al., 2016; Jang et al., 2017; Yu et al., 2019; 2024; Wang et al., 2024b) is an advanced domain in multimodal video understanding, enabling models to answer questions based on video by comprehensively interpreting both visual and textual semantics. Early works like MovieQA (Tapaswi et al., 2016) use movies as visual sources and pose questions that require understanding long temporal correspondences and dialogue logic. TGIF-QA (Jang et al., 2017) introduces more challenging question types involving repeating actions, and state transitions, necessitating spatiotemporal reasoning. Causal-VidQA (Li et al., 2022) explores commonsense and evidence reasoning. Recent NExT-GQA (Xiao et al., 2024) emphasizes the visual evidence for answers, akin to our GROUNDMORE, but we additionally provide pixel-level annotations and focus specifically on motion. PerceptionTest (Patraucean et al., 2024) is a benchmark designed to evaluate multimodal video models' perception and reasoning skills. It includes grounded video QA but lacks motion grounding at the pixel level. Our Motion-Grounded Video Reasoning is presented as a Video QA task where the answer is spatiotemporal masks, offering a more visually concrete assessment of motion understanding.

Table 2: Comparison of different video datasets. **# Obj.** indicates the number of total object categories in the dataset and **Clip Len.** means average clip length.

| Datasets | # Videos | # Expressions | Reasoning | # Masks | # Obj. | Clip Len. |
|---|---|---|---|---|---|---|
| *Video Question-Answering* | | | | | | |
| NExT-GQA (Xiao et al., 2024) | 5,417 | 43,043 | ✓ | - | - | 43.60s |
| Causal-VidQA (Li et al., 2022) | 26,900 | 107,600 | ✓ | - | - | 9.00s |
| Perception Test (Patraucean et al., 2024) | 11,620 | 38,000 | ✓ | - | 190K | 23s |
| *Action Detection* | | | | | | |
| UCF101-24 (Soomro et al., 2012) | 3,207 | - | ✗ | - | 4,458 | 6.90s |
| AVA (Gu et al., 2018) | 430 | - | ✗ | - | 56K | 15m |
| FineGym (Shao et al., 2020) | 4,883 | - | ✗ | - | 32.7K | 10m |
| MultiSports (Li et al., 2021) | 3,200 | - | ✗ | - | 37.7K | 20.9s |
| *Referring Video Segmentation / Video Grounding* | | | | | | |
| Ref-YouTube-VOS (Xu et al., 2018) | 3,978 | 15,009 | ✗ | 131K | 7,451 | 5.45s |
| Ref-Davis17 (Khoreva et al., 2019) | 90 | 1,544 | ✗ | 13.5K | 205 | 2.87s |
| MeViS (Ding et al., 2023) | 2,006 | 28,570 | ✗ | 443K | 8,171 | 13.16s |
| VidSTG (Zhang et al., 2020) | 6,924 | 99,943 | ✓ | - | 50K | 28.01s |
| *Motion-Grounded Video Reasoning (Ours)* | | | | | | |
| GROUNDMORE | 1,673 | 7,301 | ✓ | 243.6K | 3,942 | 9.61s |

## 3 GROUNDMORE FOR MOTION-GROUNDED VIDEO REASONING

### 3.1 MOTION-GROUNDED VIDEO REASONING

**Task Definition**. We propose Motion-Grounded Video Reasoning as a comprehensive motion understanding task. Basically, the input is a video clip $V \in R^{t \times h \times w \times 3}$ ($t$, $w$, $h$, 3 represent video length, width, height, and channel numbers, respectively), and a corresponding question $Q$ that is related to a specific motion, the direct answer is an object in this video clip. To let the model understand when/where the motion occurs and generate a grounded response at the pixel level, we require binary object segmentation masks $M \in R^{t' \times h \times w}$ ($t' \leq t$) related to the motion as the output.

**Task Challenges**. The key challenges of the proposed Motion-Grounded Video Reasoning lie in the following: **1) motion-related reasoning** ability towards questions and **2) pixel-level understanding** ability of the target moving object in both spatial and temporal dimensions. Concretely, for the first point, the model needs to grasp the relationship between the target motion and its spatiotemporal context, for instance, in the video where "the girl fed the dog with a piece of dog food after taking the dog food out from the cabinet". For the motion "fed", to fully understand this concept, its spatial contexts "the girl" and "a piece of dog food" should also be well perceived; and the temporal context, which is the temporally adjacent motion "taking the dog food out from the cabinet" should be understood as well since it serves as the temporal constraint on the answer. Then, based on the question "Who fed the dog with a piece of dog food after taking the dog food out from the cabinet?", only when all the spatiotemporal contexts are well grasped could the model know the answer. Second, once the model reasons about the answer, it is also required that a sequence of spatiotemporal masks represent the answer since only language output cannot avoid biased response (Xiao et al., 2024) (e.g., in a common scenario of ball game video, when asking about the motion "play", existing QA models tend to answer "balls" even without visual clues). This is of vital importance in our task, since only in a way of visual response could we know whether the model is aware of when and with what/whom the motion takes place.

### 3.2 VIDEO COLLECTION

Considering that pixel-level response is required in our Motion Grounded Video Reasoning, we carefully selected high-resolution videos (720p) from YouTube as our source videos. To ensure there are enough motion semantic and reasoning concepts in our dataset, we selected the videos from 4 scenarios: **family, animal, ball game, and outdoor activity**. Specifically, family videos usually include sufficient indoor human-human and human-object interaction, covering representative daily events such as cooking, parties, etc. Animal videos contain wild animal interactions and also a lot of human-pet interactions. Ball game videos include the most common ball-related sports such as basketball, soccer, etc. Such videos often consist of a series of intensive motions that bond with strong temporal correspondence in the players. Finally, outdoor activity videos contain general outdoor

events such as hiking, and surfing as well as normal events like kids playing in the park. We designed our dataset in this way to guarantee that it could be a benchmark with diverse video types to evaluate the comprehensive motion-related reasoning in daily life. The details of video scenes can be found in Appendix A.1. Further, we selected short clips that contain abundant motion semantics, and most of them are between 5 and 15 seconds. To ensure sufficient temporal information will be included in GROUNDMORE, we intentionally exclude samples where the motion understanding could be easily addressed without temporal information. The comparison between GROUNDMORE and other related datasets is shown in Table 2. Note that the most similar datasets are MeViS (Ding et al., 2023) and VidSTG (Zhang et al., 2020). However, MeViS does not support implicit reasoning, where the input expression contains the identity of the answer; while VidSTG focuses more on general object relation, and pixel-level annotation is not provided. More discussion on the necessity of GROUNDMORE is provided in Appendix A.4.

### 3.3 ANNOTATION PIPELINE

We recruited a team of 15 computer science students with experience in video understanding as our paid annotators to ensure high-quality annotations, 10 of them were assigned to question annotation and the rest focused on mask. For ease of the annotation, we design a **2-stage annotation** pipeline for our question annotation: 1) motion-related expression annotation; 2) LLM-assisted QA generation.

**Question Annotation Stage 1: Motion-related expression annotation.** Formally, interaction-causal expressions are with the following format: <obj_A, motion, obj_B, to do something>. Such expression could reveal the motivation behind a specific motion. Interaction-temporal expressions enable the analysis between temporally adjacent motions, which follows the format: <obj_A, motion, obj_B, before/after another motion>. In this setting, we want the model to understand motion in a temporal context and the question generated from this expression could assess the temporal awareness of the models. Moreover, we also have descriptive expression, which includes general dynamic scene descriptions and motion-related attributes that are abstracted from specific motions. The second descriptive expression could be much more challenging since it did not mention any motions here but requires detailed cross-modal and commonsense reasoning.

**Question Annotation Stage 2: LLM-assisted QA generation.** We define 4 types of questions in our GROUNDMORE dataset: **Causal** questions are generated from interaction-causal expressions, which challenge models to understand the complex relationship within interactions based on some motivations behind them. **Sequential** and **Counterfactual** questions are both generated from interaction-temporal expressions. The former investigates the chronological relations between different motions and the latter requires outstanding reasoning ability to imagine situations where it conflicts with reality. **Descriptive** questions are converted from descriptive questions. It assesses the ability to understand general scenes and use visual commonsense reasoning. Several QA examples are shown in Figure 2 and the detailed question type statistics can be found in Appendix A.1.

Before question generation, we ask our annotators to additionally annotate an index for each object related to the potential answer in our expressions in order to point out what to target in each question for the LLM we use. Basically, we leverage the strong text generation ability of GPT-4 for our question generation. We carefully design a prompt in an in-context manner (details in Appendix A.2) that requires GPT-4 to generate a question and the corresponding answer based on the expression and the target objects. The annotators manually check all of the QAs to ensure the quality.

**Mask Annotation.** We utilized the interactive tool of XMem++ (Bekuzarov et al., 2023) as our mask annotation tool. To begin with, we ask our annotators to annotate the motion timestamp for spatiotemporal mask annotation additionally. Concretely, given the video clips and the corresponding object ID information, the annotators are asked to annotate the masks for each of the objects within the motion time range. In Figure 2, we show several representative examples of our GROUNDMORE. More annotation details and more examples are provided in the Appendix A.2.

**Quality Control.** After completing the annotation process, the dataset is distributed to different annotators for quality validation. A question annotation is considered qualified if the annotator can derive the same answer as originally annotated based on the video clips. In the mask annotation, there are usually two common issues. The first is the correct mask-answer pair but poor mask quality; the second is the wrong mask-answer pair. For the first case, the annotator will improve the quality and the original annotator will check again, this process will end until the instance meets the required

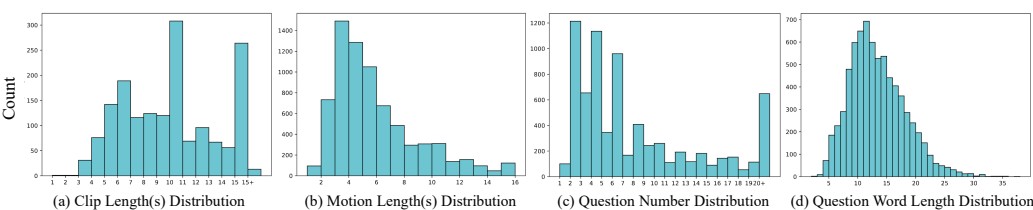

Figure 2: **Visualizations of GROUNDMORE**, including videos, questions, and visual answers (masks). Answer colors correspond to the masks. More examples are in Figure13 in Appendix.

Figure 3: Statistics of GROUNDMORE dataset.

standard; for the second case, since it will take less effort to annotate a new instance, we just directly discard those defective annotations. In the end, all of the mask-answer pairs will meet the criteria. More details can be found in Appendix A.2.

## 3.4 DATASET STATISTICS

We compare our GROUNDMORE with existing popular RVOS datasets Ref-YouTube-VOS (Seo et al., 2020), Ref-Davis17 (Khoreva et al., 2019), and the recent MeViS (Ding et al., 2023). Our GROUNDMORE contains 1,673 videos 7,301 questions and 243K object masks as well as 3,942 objects. And the average video clip duration is 9.61 seconds. GROUNDMORE is split into 800 training, 150 validation, and 723 test videos, roughly a 50:10:40 split. Following (Lai et al., 2023), we intentionally reduce the scale of the training split due to the strong zero-shot ability of current multimodal LLMs, and we ensure there are sufficient test samples for persuasive benchmarking.

As shown in Figure 3a, most of the clips have a duration between 5s and 15s, which is long enough to include sufficient motion semantics. This range ensures that the clips capture complete actions and interactions, providing a rich context for question formulation. In Figure 3b, it is evident that most motions in GROUNDMORE have a duration from 2s to 6s, highlighting the challenge of temporal localization in our dataset. These short-duration motions require precise temporal understanding and segmentation, adding to the complexity of the GROUNDMORE. Besides, the average motion (segment) ratio in each video clip is 51%. As seen in Figure 3c, for most clips, the number of questions is more than 2, with a significant number having up to 4 or more questions. This indicates that GROUNDMORE provides a diverse set of questions per clip, ensuring a comprehensive evaluation of the clip's content. It also implies that each clip contains multiple distinct motion semantics that warrants varied questioning. In Figure 3d, the distribution shows that most questions are sufficiently long, typically ranging from 7 to 15 words. This length reflects the complexity and detail required in the questions, underscoring the difficulty level of our GROUNDMORE. The substantial word count in questions ensures that they are descriptive and context-rich, further challenging the systems to provide accurate and detailed responses. More details including figures of statistics are in the Appendix A.1.

## 4 EXPERIMENTS

In this section, we first list popular image/video grounding frameworks (Sec. 4.1). Then we introduce our proposed baseline **Mo**tion-Grounded Video **R**easoning **A**ssistant (MoRA) (Sec. 4.2). Next, we provide detailed evaluation results and analysis in terms of reasoning ability, temporal context, and the localization branch (Sec. 4.3).

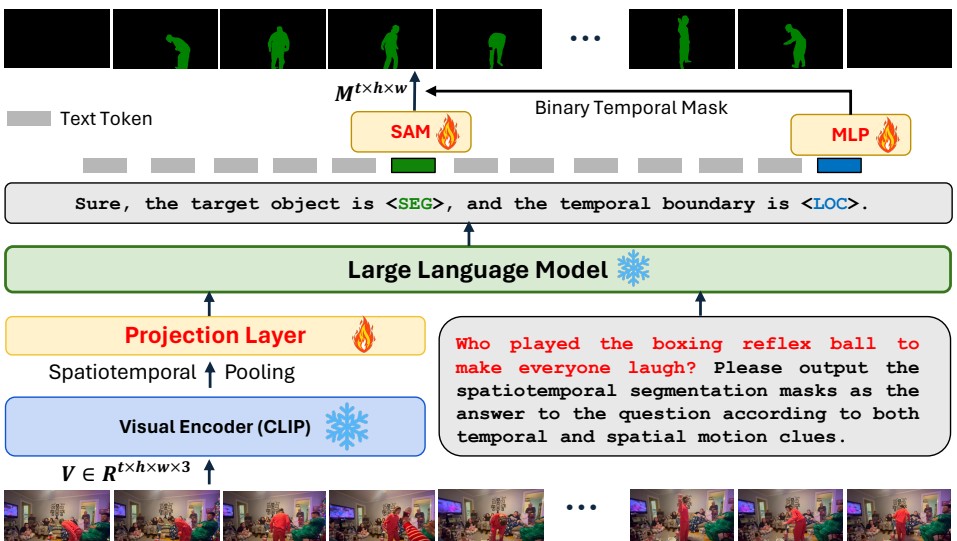

Figure 4: MoRA adopts the spatiotemporal pooling strategy and inserts the extra special **[SEG]** token. Additionally, to enable the temporal localization ability, MoRA takes advantage of the extra **[LOC]** token to learn a binary temporal mask, which refines the direct SAM outputs.

## 4.1 BASELINE MODELS FOR EVALUATION

We choose baselines including **1) Referring VOS Models**: ReferFormer (Wu et al., 2022b), SgMg (Miao et al., 2023), HTR (Miao et al., 2024), and LMPM (Ding et al., 2023), that are pure visual segmentation models and without LLMs. **2) Image Reasoning Segmentation Models**: LISA (Lai et al., 2023) and PixelLM (Zhongwei et al., 2023) that have strong LLM and are equipped with extra spatial grounding heads. We adapt them to videos in a frame-by-frame manner. **3) Video Reasoning Segmentation Models:** PG-Video-LLaVA (Munasinghe et al., 2023) that is build upon video-LLM (Maaz et al., 2023) and strong grounding modules (Kirillov et al., 2023; Liu et al., 2023b; Cheng et al., 2023a). Since our task could be solved in a non-end-to-end, two-stage manner (answering first, segmentation next), we also evaluate **4) Two-stage Baselines** that are composed by strong video QA models (ViLA (Lin et al., 2024), VideoChat2 (Li et al., 2024) and SeViLA (Yu et al., 2023)) and Referring VOS models.

## 4.2 OUR METHOD: MOTION-GROUNDED VIDEO REASONING ASSISTANT

Our **Mo**tion-Grounded Video **R**easoning **A**ssistant (MoRA) is built upon LISA (Lai et al., 2023), which is an image-based reasoning segmentation framework, equipping the strong LLaVA (Liu et al., 2023a) and SAM (Kirillov et al., 2023). To perform an efficient frame encoding, we take advantage of the spatiotemporal pooling mechanism in Video-ChatGPT (Maaz et al., 2023). We leverage the segmentation token **[SEG]** in LISA for spatial segmentation. However, one of the most challenging points in our task is that we need not only to segment the objects in the spatial dimension but also to localize them temporally. Therefore, as shown in Figure 4, to construct a unified LLM-based framework, we leverage extra **[LOC]** tokens to encode the temporal boundary information in the language space. The **[LOC]** embedding will be decoded by an MLP layer into a temporal mask to prevent false activations during frame-wise mask decoding.

In training, we directly initialize our MoRA with a pre-trained LISA due to its well-leaned text-object alignment. Further, in order to adapt the model with vision-language alignment in the video domain, we first pre-train it with the Ref-YouTubeVOS (Xu et al., 2018) and MeViS (Ding et al., 2023) dataset (we convert the original text annotation into QA formats to force MORA to follow the instructions) for 20 epochs without the temporal localization module, which could be used for zero-shot evaluation. Further, we finetune MoRA, equipped with the localization module, with the training split of GROUNDMORE for another 20 epochs.

## 4.3 EVALUATION AND ANALYSIS

**Metrics.** Following prior works (Khoreva et al., 2019; Seo et al., 2020; Ding et al., 2023), we use the popular metrics: Jaccard index ($\mathcal{J}$) (Jaccard, 1912) and F-measure ($\mathcal{F}$) (Dice, 1945). $\mathcal{J}$ estimates the IoU of the predicted and the GT masks, $\mathcal{F}$ indicates contour accuracy. We also report $\mathcal{J}\&\mathcal{F}$ to reflect overall performance. We evaluate models on GROUNDMORE across question types, revealing their grounding and reasoning ability from different aspects.

**Baseline Comparisons.** As shown in Table 3, we first replace the questions with the titles of the corresponding YouTube videos and run as an RVOS task with noisy text labels using ReferFormer (Wu et al., 2022b) as the random baseline. Compared with the random baselines, RVOS models achieve reasonable improvements, especially LMPM (Ding et al., 2023), which is also trained by MeViS (Ding et al., 2023) data that contains more motion-related data than simple referring VOS datasets (Seo et al., 2020; Khoreva et al., 2019). Surprisingly, image reasoning segmentation baselines (Lai et al., 2023; Zhongwei et al., 2023), with strong LLM, are lower than RVOS models. The reason could be the lack of temporal modeling in those image-level models, which makes it hard to propagate target object information across frames. For PG-Video-LLaVA (Munasinghe et al., 2023), though it is a video reasoning segmentation/grounding model, the performance is not even higher than the best RVOS model. A potential reason could be that it tends to ground all salient objects given the scene description due to the redundant response of its video LLM (Maaz et al., 2023), resulting in more false positives. For two-stage baselines, we could also observe superior performance over PG-Video-LLaVA. Comparing the video LLM in PG-Video-LLaVA and the other three (Yu et al., 2023; Lin et al., 2024; Li et al., 2024), the most important reason is that Video-ChatGPT tends to generate overlong answers, which could be ambiguous for grounding models to locate target objects. Details of the video LLMs in the two-stage baselines can be found in Appendix A.6. For different question types, we can also observe that in *Causal* and *Descriptive* questions, two-stage baselines built upon ViLA and SeViLA perform better than MORA, we hypothesize that ViLA and SeViLA maintain their strong reasoning ability in these two types of questions when not trained with an additional grounding module; while in the temporal-related questions (i.e., *Sequential* and *Counterfactual*), the temporal head in our MORA makes a difference.

Conclusively, our MORA achieves new state-of-the-art, outperforming the best existing video reasoning grounding model (PG-Video-LLaVA) by an average of 28.8% relatively. The reasons could be two-fold: (1) the language model in PG-Video-LLaVA provides ambiguous response for its grounding modules while the **[SEG]** token in MORA is trained in an end-to-end manner, conveying more informative features of target objects; (2) PG-Video-LLaVA, as well as other baselines, does not include any temporal localization design while the **[LOC]** in MORA, supervised by the timestamps of the motion, could lead to accurate temporal estimation.

However, the design of our MORA is still basic and there is substantial room for future improvements in both model training and model design. For instance, the LLaVA could be replaced with better LLMs which are trained with more motion-sensitive language corpus to enhance visual-language alignment in dynamic scenes; the spatiotemporal pooling, though efficient, could inevitably cause information loss; and better time-sensitive modeling could also replace the simple temporal localization head.

**Dataset Diagnosis.** In order to showcase that our GROUNDMORE indeed introduces challenges mentioned in Sec. 3.1, we diagnose GROUNDMORE from two aspects, implicit reasoning and temporal context. We examine implicit reasoning by comparing the evaluation metrics between the original setting and replacing questions with the ground truth answer, which could be viewed as referring to spatiotemporal video segmentation. As shown in Table 4, providing GT answers could largely alleviate the difficulty of the task, resulting in an average of 55.93% relative improvement in $\mathcal{J}\&\mathcal{F}$. For temporal context diagnosis, we simply leverage the temporal annotation of the spatiotemporal masks to segment the original clip and input these motion-heavy clips into the models. The tasks are easier without temporal context since only spatial grounding is required. As shown in Table 4, comparing the first row and the third row for each model, we could observe a relative improvement of 48.96%. This diagnosis indicates that the QA design and the temporal localization feature contribute a lot to its challenge.

**Temporal Localization Branch.** For ablation, we further fine-tune our MoRA with or without the temporal localization branch, as shown in Table 5. This branch brings an 8.7% relative boost, and for all but *Descriptive* questions the improvements are obvious, indicating the localization is important

Table 3: **Motion-Grounded Video Reasoning results** on our GROUNDMORE. We compare all methods in a zero-shot setting. We **bold** the best numbers, and underlined the second-best numbers.

| Methods | Overall | | | Causal | | | Sequential | | | Counterfactual | | | Descriptive | | |
|---|---|---|---|---|---|---|---|---|---|---|---|---|---|---|---|
| | J&F | J | F | J&F | J | F | J&F | J | F | J&F | J | F | J&F | J | F |
| *Random Baseline* | | | | | | | | | | | | | | | |
| Title+ReferFormer (Wu et al., 2022b) | 9.89 | 9.78 | 10.00 | 9.63 | 9.40 | 9.85 | 9.32 | 9.20 | 9.43 | 9.22 | 9.11 | 9.32 | 10.89 | 10.89 | 10.88 |
| *RVOS Baseline* | | | | | | | | | | | | | | | |
| ReferFormer (Wu et al., 2022b) | 10.71 | 10.75 | 10.68 | 9.88 | 9.79 | 9.98 | 9.41 | 9.39 | 9.44 | 11.02 | 10.99 | 11.06 | 12.14 | 12.35 | 11.93 |
| SgMg (Miao et al., 2023) | 12.55 | 12.82 | 12.28 | 12.10 | 12.23 | 11.97 | 11.16 | 11.35 | 10.97 | 13.59 | 13.74 | 13.44 | 13.26 | 13.79 | 12.73 |
| HTR (Miao et al., 2024) | 10.41 | 10.34 | 10.48 | 10.13 | 9.96 | 10.30 | 9.22 | 9.09 | 9.34 | 10.42 | 10.29 | 10.54 | 11.42 | 11.51 | 11.32 |
| LMPM (Ding et al., 2023) | 12.97 | 13.04 | 12.90 | 11.89 | 12.31 | 11.47 | 11.04 | 11.17 | 10.91 | 13.17 | 13.18 | 13.19 | 12.76 | 12.56 | 12.96 |
| *Image Reasoning Segmentation Baseline* | | | | | | | | | | | | | | | |
| LISA-7B (Lai et al., 2023) | 8.01 | 8.29 | 7.83 | 7.55 | 7.45 | 7.65 | 7.79 | 8.03 | 7.55 | 6.77 | 6.48 | 7.06 | 9.44 | 10.01 | 8.88 |
| LISA-13B (Lai et al., 2023) | 8.24 | 8.80 | 7.67 | 7.09 | 7.85 | 6.33 | 7.81 | 8.17 | 7.46 | 7.28 | 7.61 | 6.94 | 10.48 | 11.35 | 9.61 |
| PixelLM-7B (Zhongwei et al., 2023) | 9.38 | 9.49 | 9.27 | 9.11 | 9.21 | 9.01 | 9.01 | 9.23 | 8.79 | 10.44 | 10.87 | 10.01 | 9.95 | 10.01 | 9.89 |
| PixelLM-13B (Zhongwei et al., 2023) | 11.24 | 11.00 | 11.48 | 10.96 | 11.07 | 10.85 | 12.04 | 12.18 | 11.90 | 10.40 | 10.85 | 9.95 | 11.37 | 12.56 | 11.18 |
| *Video Reasoning Segmentation Baseline* | | | | | | | | | | | | | | | |
| PG-Video-LLaVA (Munasinghe et al., 2023) | 11.96 | 11.35 | 12.57 | 10.48 | 10.24 | 10.72 | 11.75 | 12.76 | 10.74 | 12.18 | 12.01 | 12.36 | 12.45 | 13.21 | 11.69 |
| PG-Video-LLaVA+SAM2 (Ravi et al., 2024) | 11.88 | 11.32 | 12.44 | 10.94 | 11.21 | 10.67 | 12.05 | 11.98 | 12.12 | 12.30 | 12.40 | 12.20 | 12.33 | 10.17 | 14.49 |
| *Two-Stage Baseline* | | | | | | | | | | | | | | | |
| ViLA (Lin et al., 2024)+ReferFormer | 13.92 | 12.61 | 15.23 | 12.92 | 11.59 | 14.24 | 13.40 | 12.32 | 14.48 | 11.56 | 10.48 | 12.64 | 16.56 | 14.98 | **18.14** |
| ViLA (Lin et al., 2024)+SgMg | 13.87 | 12.44 | 15.29 | 12.79 | 11.32 | 14.26 | 13.18 | 11.91 | 14.44 | 12.47 | 11.37 | 13.56 | 16.12 | 14.44 | 17.80 |
| ViLA (Lin et al., 2024)+HTR | 13.34 | 11.90 | 14.77 | 12.60 | 11.17 | 14.03 | 12.53 | 11.28 | 13.78 | 11.35 | 10.18 | 12.52 | 15.68 | 13.99 | 17.37 |
| VideoChat2 (Li et al., 2024)+ReferFormer | 13.06 | 11.68 | 14.44 | 12.41 | 11.02 | 13.79 | 12.44 | 11.33 | 13.54 | 10.61 | 9.48 | 11.73 | 15.47 | 13.78 | 17.16 |
| VideoChat2 (Li et al., 2024)+SgMg | 13.23 | 11.76 | 14.70 | 12.50 | 11.02 | 13.98 | 12.60 | 11.32 | 13.88 | 11.92 | 10.76 | 13.08 | 15.07 | 13.31 | 16.82 |
| VideoChat2 (Li et al., 2024)+HTR | 12.61 | 11.13 | 14.09 | 12.37 | 10.89 | 13.84 | 11.94 | 10.67 | 13.21 | 10.87 | 9.64 | 12.10 | 14.26 | 12.49 | 16.02 |
| SeViLA (Yu et al., 2023)+ReferFormer | 13.77 | 14.50 | 13.03 | 13.56 | 14.13 | 12.98 | 12.20 | 12.70 | 11.70 | 11.49 | 11.95 | 11.03 | 16.23 | 17.41 | 15.04 |
| SeViLA (Yu et al., 2023)+SgMg | 15.30 | **16.04** | 14.56 | **15.81** | **16.46** | **15.17** | 13.77 | 14.21 | 13.33 | 13.20 | 13.73 | 12.67 | **16.94** | **18.08** | 15.81 |
| SeViLA (Yu et al., 2023)+HTR | 13.91 | 14.60 | 13.23 | 13.81 | 14.36 | 13.25 | 12.43 | 12.89 | 11.97 | 12.08 | 12.55 | 11.61 | 15.98 | 17.04 | 14.92 |
| MoRA (Ours) | **15.53** | 15.46 | **15.60** | 14.26 | 14.08 | 14.45 | **14.70** | **14.56** | **14.84** | **17.51** | **17.08** | **17.94** | 16.15 | 16.45 | 15.84 |

Table 4: **Dataset diagnostics** w.r.t. implicit reasoning and temporal context.

| Methods | Implict Reasoning | Temporal Context | Overall | | | Causal | | | Sequential | | | Counterfactual | | | Descriptive | | |
|---|---|---|---|---|---|---|---|---|---|---|---|---|---|---|---|---|---|
| | | | J&F | J | F | J&F | J | F | J&F | J | F | J&F | J | F | J&F | J | F |
| ReferFormer | ✓ | ✓ | 10.71 | 10.75 | 10.68 | 9.88 | 9.79 | 9.98 | 9.41 | 9.39 | 9.44 | 11.02 | 10.99 | 11.06 | 12.14 | 12.35 | 11.93 |
| | ✗ | ✓ | 16.37 | 14.97 | 17.78 | 14.63 | 13.22 | 16.03 | 12.89 | 11.99 | 13.79 | 13.17 | 12.39 | 13.96 | 22.02 | 19.94 | 24.10 |
| | ✓ | ✗ | 17.03 | 18.01 | 16.04 | 16.30 | 17.07 | 15.53 | 15.43 | 16.16 | 14.69 | 15.57 | 16.29 | 14.86 | 19.53 | 21.03 | 18.04 |
| SgMg | ✓ | ✓ | 12.55 | 12.82 | 12.28 | 12.10 | 12.23 | 11.97 | 11.16 | 11.35 | 10.97 | 13.59 | 13.74 | 13.44 | 13.26 | 13.79 | 12.73 |
| | ✗ | ✓ | 19.15 | 17.83 | 20.47 | 18.68 | 17.23 | 20.12 | 15.61 | 14.68 | 16.53 | 16.07 | 15.30 | 16.84 | 23.52 | 21.77 | 25.28 |
| | ✓ | ✗ | 16.84 | 17.79 | 15.89 | 16.79 | 17.59 | 15.99 | 15.05 | 15.76 | 14.34 | 14.98 | 15.66 | 14.31 | 19.05 | 20.43 | 17.66 |
| HTR | ✓ | ✓ | 10.41 | 10.34 | 10.48 | 10.13 | 9.96 | 10.30 | 9.22 | 9.09 | 9.34 | 10.42 | 10.29 | 10.54 | 11.42 | 11.51 | 11.32 |
| | ✗ | ✓ | 16.90 | 15.31 | 18.49 | 16.18 | 14.57 | 17.78 | 13.12 | 11.88 | 14.35 | 13.63 | 12.61 | 14.65 | 21.79 | 19.67 | 23.91 |
| | ✓ | ✗ | 16.00 | 16.87 | 15.13 | 15.67 | 16.41 | 14.92 | 14.50 | 15.15 | 13.86 | 14.61 | 15.22 | 13.99 | 18.03 | 19.31 | 16.75 |

in the other three questions, which is consistent with the conclusion in reasoning ability analysis in Table 4. Besides, we can observe that, without the temporal localization branch, fine-tuning can still bring an obvious improvement, especially for *Causal* and *Descriptive* questions, indicating that for the rest two types, weak temporal awareness could impair the performance gain from additional data.

Table 5: **Ablation studies of the localization branch** in MORA. zs: zero-shot, ft: fine-tuned.

| Methods | Overall | | | Causal | | | Sequential | | | Counterfactual | | | Descriptive | | |
|---|---|---|---|---|---|---|---|---|---|---|---|---|---|---|---|
| | J&F | J | F | J&F | J | F | J&F | J | F | J&F | J | F | J&F | J | F |
| MoRA-zs | 15.53 | 15.46 | 15.60 | 14.26 | 14.08 | 14.45 | 14.70 | 14.56 | 14.84 | 17.51 | 17.08 | 17.94 | 16.15 | 16.45 | 15.84 |
| MoRA-ft w/o loc. | 18.14 | 18.52 | 17.86 | 18.71 | 18.03 | 19.38 | 17.23 | 17.59 | 16.87 | 17.08 | 17.29 | 16.86 | 20.88 | 20.94 | 20.82 |
| MoRA-ft | **19.72** | **19.52** | **19.92** | **20.21** | **20.02** | **20.40** | **19.03** | **19.88** | **18.18** | **18.66** | **18.45** | **18.87** | **21.69** | **22.03** | **21.35** |

## 5 CONCLUSION

In this paper, we propose a new video task called Motion-Grounded Video Reasoning for comprehensive motion understanding. We consider motion as a combination of its spatiotemporal contexts and design QA to force models to understand implicit textual input and thus reason about the motion-related objects. Further, we point out that due to the spatiotemporal nature of motion, solely output text answers could be vague, which cannot directly illustrate when and where a specific motion takes place. Considering this, we design to output spatiotemporal masks of motion-related objects, which is a direct and explainable way to address the issue. To meet the evaluation requirement, we also collect a large-scale dataset called GROUNDMORE, which includes 4 types of questions that could evaluate different aspects of motion reasoning abilities. Finally, our simple baseline, MORA, achieved reasonable performance on the new dataset, but the low score compared to other video datasets reveals that there is still much to explore for motion reasoning and understanding. The limitation of our work can be found in Appendix A.7.

**Ethics Statement.** Our GROUNDMORE is constructed from publicly available videos on YouTube, where all sourced videos are licensed under the Creative Commons License. The dataset consists of segments or clips from the original videos, rather than full-length videos, and has been annotated by our annotator group. The dataset is intended exclusively for non-commercial research and educational purposes. In accordance with ethical guidelines, researchers using this dataset are expected to prioritize privacy, fairness, and the ethical use of data when analyzing or disseminating findings based on GROUNDMORE. Any potential misuse of the dataset, including re-identification or other actions that may harm individuals depicted in the videos, is strictly prohibited. By using GROUNDMORE, researchers agree to adhere to these privacy and ethical standards. Please see Appendix A.8 for more details about copyright and privacy statements.

**Reproducibility Statement.** The dataset (we have attached part of the dataset in the Supplementary Materials, the full version will be released after acceptance), code, and model will be open-sourced. Moreover, the training detail of our baseline model MORA is described in Sec 4.2.

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

# A APPENDIX

The following appendix is structured to provide supplementary information about our GROUNDMORE dataset, its annotations, and representative examples. We aim to present a comprehensive view of the statistical analysis, annotation process, and key insights that further elaborate on the main text. The appendix is divided into the following sections:

- Section A.1 offers detailed statistical insights into the types of questions and scenes captured in our dataset, as well as an analysis of the distribution of objects, verbs, and word clouds in the question annotations.

- Section A.2 provides detailed information about the annotation process, including the types of motion-related expressions, the generation of questions through large language models, and the quality validation procedures.

- Section A.3 showcases a set of representative examples from GROUNDMORE, illustrating the richness of the dataset through diverse scenes, objects, and questions.

- Section A.4 discusses the necessity of including implicit reasoning, highlighting the importance of capturing nuanced motion-grounded video reasoning.

- Section A.5 showcases the impact of object numbers on the dataset's performance.

- Section A.6 demonstrates the qualitative performance of current video LLMs in the two-stage baseline settings.

- Section A.7 outlines the limitations of the current version of GROUNDMORE and discusses future work.

- Section A.8 outlines the ethical considerations, privacy concerns, and licensing terms associated with GROUNDMORE.

## A.1 GROUNDMORE STATISTICS

**A.1.1 Question and Scene Type.** We provide detailed statistics of GROUNDMORE in this section, including the distribution of question types, scene types, objects, and verbs that appear in our question annotation, etc. As shown in Figure 5a, the **Descriptive** questions constitute the highest proportion at 29.7%, followed closely by **Causal** questions at 28.5%. **Sequential** questions make up 21.7% of the total, while **Counterfactual** questions are the least common, accounting for 20.2%. Our GROUNDMORE shows a balanced distribution w.r.t. question type. Regarding scene type distribution (Figure 5b), **family** scenes dominate with a significant 35.1% share, slightly higher than the **ball game** scenes, which account for 32.7%. **Animal** scenes are also well-represented at 25.4%, whereas **outdoor activity** scenes are relatively rare, comprising only 6.8% of the total scenes in our GROUNDMORE.

**A.1.2 Object Word Distribution.** Figure 6 illustrates the top 30 most frequent objects in our GROUNDMORE questions. We categorize these objects into six parent categories: sports equipment, people, animals, furniture, household items, and food, reflecting common items in daily life. As can be seen from the figure, *ball* is the most frequently occurring object, followed by *man*, *dog*, *basketball*, and *girl*. This prevalence is aligned with the high proportion of sports and family videos in our GROUNDMORE, as indicated in Figure 5b. The dominance of sports equipment such as *ball* and *basketball* correlates with the 32.7% share of ball game scenes. Similarly, the frequent appearance of *man*, *girl*, and *woman* objects is consistent with the substantial 35.1% of family scenes, where people are commonly depicted. Additionally, animals like *dog* and *cat* are prominent due to their significant 25.4% representation in animal scenes. The distribution of these objects highlights the diverse and realistic contexts covered in our GROUNDMORE, ensuring a comprehensive evaluation of various question types and scene contexts.

**A.1.3 Verb Distribution.** Another key component of our GROUNDMORE is the verb in the motion-related questions. In Figure 7, we present the top 20 most frequent verbs across different scene types,

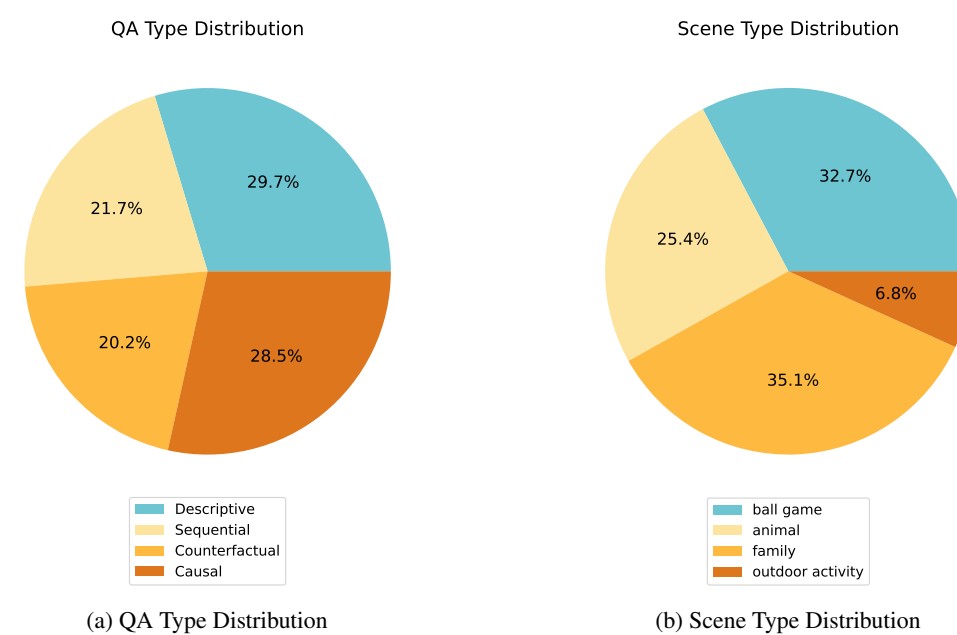

(a) QA Type Distribution        (b) Scene Type Distribution

Figure 5: Question and Scene Type Distribution of GROUNDMORE.

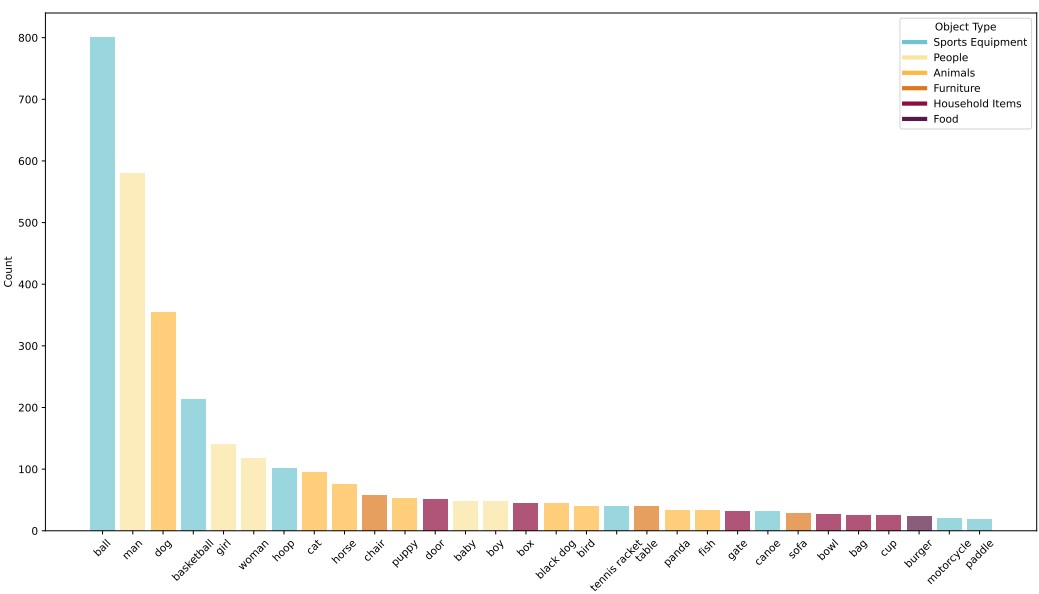

Figure 6: Object distribution of GROUNDMORE.

represented by distinct colors. The verb *use* has the highest overall proportion, reflecting its ubiquity in daily activities, with a notable presence in family scenes, as well as significant occurrences in animal and ball game scenes. The verb *dribble* ranks second and is exclusively found in ball game videos, highlighting its specificity to that context. The verb *move* is also prominent, appearing across all four scene types, indicating its general applicability in various contexts. Verbs such as *hold*, *open*, and *put* are more frequently observed in family videos, underscoring their relevance to everyday domestic activities. In contrast, *accelerate* and *shoot* are predominantly associated with ball game scenes, which is consistent with the dynamic nature of these activities. Besides, the distribution of verbs shows a more balanced pattern compared to the object distribution, reflecting a diverse range of actions across different contexts. For instance, while *throw* and *pass* are mainly seen in ball game scenes, verbs like *push* and *grab* are well-represented in both family and ball game contexts. This

918
919
920
921
922
923
924
925
926
927
928
929
930
931
932
933

balanced distribution underscores the comprehensive nature of our GROUNDMORE, capturing a wide array of activities and interactions within various scene types.

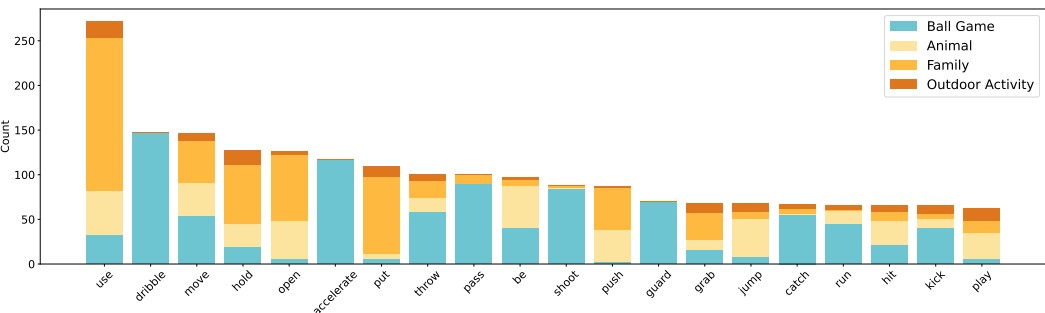

Figure 7: Verb distribution of the motion concepts in GROUNDMORE.

**A.1.4 Word Cloud Visualization.** Moreover, we leverage the word cloud of the top 100 words that appear in our GROUNDMORE questions. The word cloud in Figure 8 provides a visual representation of the most frequently occurring words. We can observe that common objects like *"dog"*, *"cat"*, and *"ball"* are prominently featured, which aligns with the object distribution shown in Figure 6. These objects are integral to many of the scenes and questions, reflecting their high frequency in the dataset. In addition to objects, prepositions closely related to motion, such as *"down"*, *"out"*, and *"with"*, are also prevalent. This is consistent with the verb distribution illustrated in Figure 7, where actions often involve directional or positional changes, necessitating the use of these prepositions. Furthermore, adverbs such as *"before"* and *"after"* appear frequently, indicating their importance in describing temporal relationships within the scenes. These temporal adverbs are essential in forming questions related to sequences and causality, which are common in descriptive and sequential question types. Overall, the word cloud highlights the interconnected nature of objects, verbs, and descriptive language within our GROUNDMORE, demonstrating the comprehensive coverage of various elements that contribute to the complexity and richness of the dataset.

**A.1.5 Sankey Diagram for Interaction.** We provide the Sankey diagram of our proposed GROUND-MORE in Figure 9, which illustrates the interactions within our GROUNDMORE. In this diagram, the elements on the left side represent different initial categories of objects or entities involved in interactions (e.g., People_A, Animals_A, Sport Equipments_A), while the elements on the right side represent the resulting categories of objects or entities after interactions (e.g., People_B, Animals_B, Sport Equipments_B). From the diagram, we can see that human-involved interactions (People_A) have the highest proportions, flowing prominently into both sports and family categories on the right. This is consistent with the scene type distribution (Figure 5b), where sports and family scenes were among the most prevalent. Similarly, the frequent appearance of sports equipment, animals, and household items in both left and right categories aligns with the object distribution shown in Figure 6. The Sankey diagram validates that our GROUNDMORE is well-suited for motion and interaction understanding. It demonstrates the comprehensive coverage of various interactions, emphasizing the importance of human involvement and the diverse range of objects and entities engaged in these interactions. This rich interplay of elements ensures that GROUNDMORE could serve as a robust benchmark for evaluating motion understanding in complex video scenarios.

## A.2 ANNOTATION DETAILS

As mentioned in Section 3.3, the question annotation is constituted of two stages: 1) motion-related expression annotation; and 2) LLM-assisted QA generation. And we resort to XMem++ (Bekuzarov et al., 2023) as our semi-automated mask annotation tool. The interface is shown in Figure 10.

**A.2.1 Expression Annotations.** Expression annotation is to annotate the ongoing motions or events in a given video. We define three different expression types: interaction-causal, interaction-temporal, and descriptive expression. The motions that can be described within these three types of expressions could generally cover most of the daily scenarios. The interaction-causal expression has the format <obj_A, motion, obj_B, to do something> which depicts a scene where the motion takes place based on some hidden motivations. For instance, as shown in the first row in Figure 11, the causal-driven

972
973
974
975
976
977
978
979
980
981
982
983
984
985
986
987
988
989
990
991
992
993
994
995
996
997
998
999
1000
1001
1002
1003
1004
1005
1006
1007
1008
1009
1010
1011
1012
1013
1014
1015
1016
1017
1018
1019
1020
1021
1022
1023
1024
1025

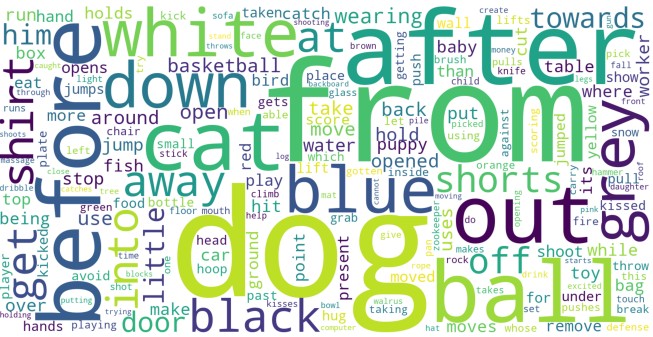

Figure 8: Word cloud of the top 100 words in the question annotation in our GROUNDMORE dataset.

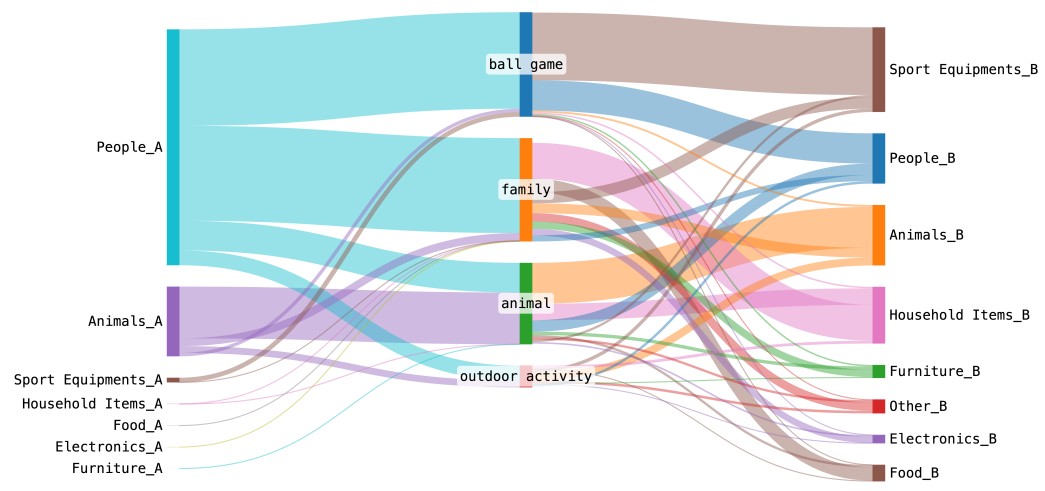

Figure 9: Sankey diagram on the interaction of our GROUNDMORE.

expression of this case elucidates the motivation behind the motion of *passed the knife to the man in the grey shirt* is to *let him cut the watermelon*. Interaction-temporal expressions, following the format <obj_A, motion, obj_B, before/after another motion>, describe the chronological relations between temporally adjacent actions, which enables motion understanding based on temporal conditions. As shown in the second row in Figure 11, *the man in black* performs two consecutive actions, *get rid of the defense from the man in white* and *shot the basketball*. In most similar cases, the temporally adjacent motion not only has temporal relations but also has cause-and-effect; therefore, such expressions could help analyze the existence of one motion based on another. The third one is the descriptive expression, which contains either general scene description or motion-based abstract attributes (e.g., *energetic, naughty, faster, etc.*). As shown in the last row in Figure 11, *consumed more energy* could be viewed as an abstract attribute represented by the fact that the man is doing massage for the dog. Given this expression type, the models are required to perform both spatiotemporal reasoning and commonsense reasoning to understand the scene content.

**A.2.2 Question Annotations.** As shown in Figure 12, we specifically design the prompt to leverage the text generation ability of GPT-4o. For each expression, we first specify the target objects that would be annotated during the mask annotation. For instance, in the first row of Figure 11, considering the bidirectional nature of an interaction, we will ask GPT-4o to generate questions for both *the man in the yellow shirt* and *the knife* by providing their object ID: {"1": "the man in the yellow shirt", "2": "the knife"}.

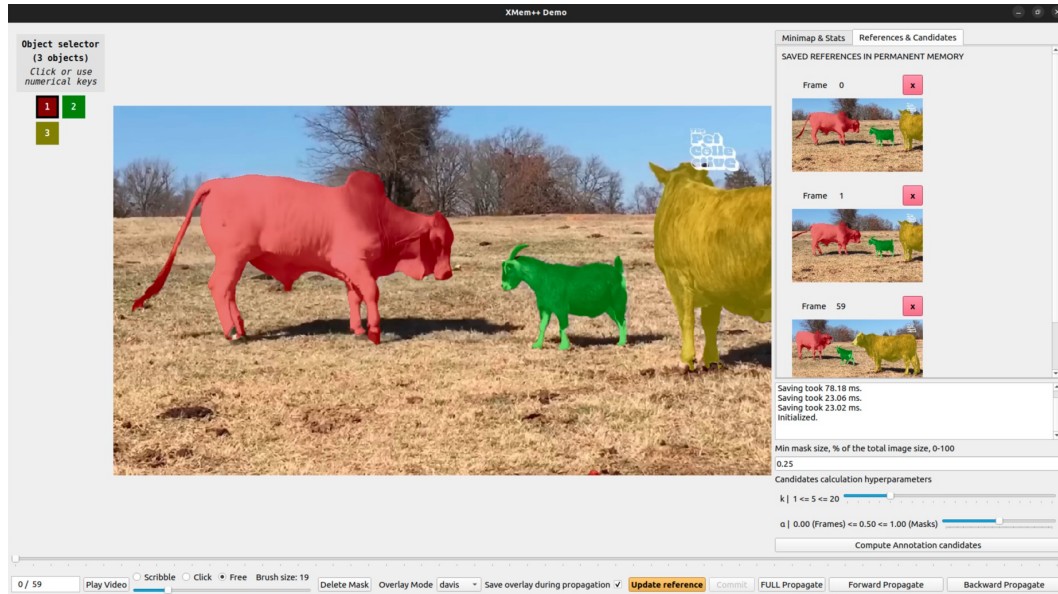

Figure 10: Annotation Interface of XMem++.

**Causal** questions are generated from expressions of interaction-causal expressions. Due to the bidirectional nature of the interactions, we will generate questions targeting both subject and object. For instance, for the expressions in the first row of Figure 11, we will generate questions as follows: *Who passed the knife to the man in the grey shirt to let him cut the watermelon?* and *What did the man in the yellow shirt pass to the man in the grey shirt to let him cut the watermelon?* We generate questions for both the subject and the object of motion to ensure complete spatial context reasoning. **Sequential** questions are generated from interaction-temporal expressions. Similarly, since it is also interaction-related, we will generate two questions for each expression as shown in the middle row of Figure 11. **Counterfactual** questions are also generated from interaction-temporal expressions. But it focuses on those scenarios where temporal-adjacent motions have cause-and-effect. For example, in the middle row of Figure 11, the fact that **the man in black got rid of the defense from the man in white** is a prerequisite that he could perform a jump shot. Therefore, the questions can be as follows: *Who needs to be got rid of defense from by the man in black or he cannot shoot the basketball?* and *What cannot be shot if the man in black did not get rid of the defense from the man in white?* **Descriptive** questions are simply converted from descriptive questions as shown in the third row of Figure 11. It will follow the same rule aforementioned if an interaction is involved.

**A.2.3 Quality Validation.** After the generation of questions by GPT-4o, the resulting questions and their corresponding answers will be distributed to different annotators in the same question annotation group for quality control. Importantly, these annotators will not have been involved in the original expression annotation to ensure objectivity. The annotators will be instructed to perform the following steps:

1. **Check relevance**: Verify whether the generated question logically aligns with the current video context and scene.

2. **Answer validation**: Answer the question independently and compare the response with the original annotation. The goal is to ensure consistency between the generated answer and the initial annotation.

3. **Single-object validation**: Confirm that the answer references a single object when appropriate. If the answer references multiple objects and is not explicitly required, the question-answer pair should be revised.

If any issues are identified with the question or the answer, the annotator is required to update the question-answer pair. For example, if the generated question is *"Who is playing baseball?"* and the answer is *"The boy and the dog"*, the annotator should revise the pair to better reflect clarity and

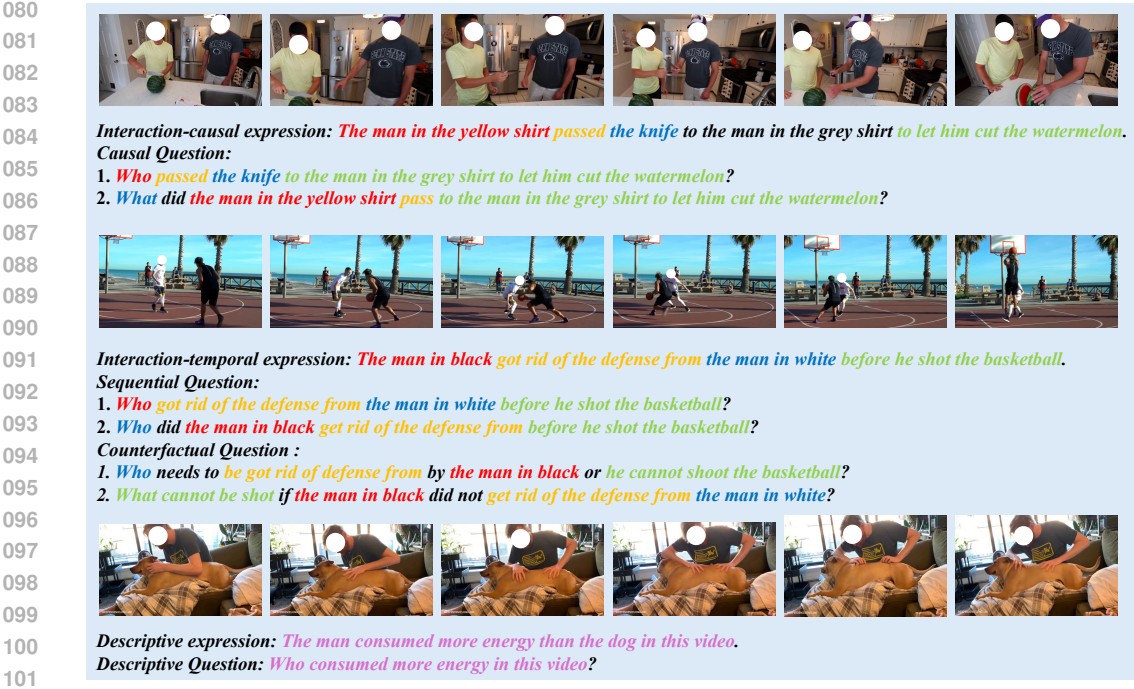

Figure 11: Question generation examples for different types of motion-related expressions.

Figure 12: QA generation prompt.

context, such as: *"Who is playing baseball with the dog? The boy."* and *"Who is playing baseball with the boy? The dog."*

Similarly, the masks will undergo a quality check by different annotators within the mask annotation group. The first task for the reviewer is to assess whether the mask corresponds to the object(s) indicated in the answer. If a mismatch is found between the mask and the answer, a third annotator will be consulted to provide an additional opinion. The final decision on whether to accept or reject the mask will be based on the majority decision. Mismatched masks will be discarded entirely since re-annotating from scratch is typically more efficient than attempting to fix them.

If the masks match the answer, the annotator will proceed to evaluate the overall quality, focusing on any potential missing regions, incorrect regions, or other inaccuracies. In the end, all mask-answer pairs must meet predefined quality standards to ensure their validity for downstream tasks.

**A.2.4 Annotator Compensation.** We compensated the question annotators $0.50 per expression and paid $1.00 per clip for mask annotations. Additionally, during the quality validation process, we provided an extra compensation of $0.20 per instance (a question-clip pair).

## A.3 GROUNDMORE EXAMPLES

We provide additional visualizations of our proposed GROUNDMORE in Figure 13. As shown, our GROUNDMORE requires advanced motion reasoning abilities in diverse scenarios. As illustrated in the fourth row of the figure, the question "What might not be held by the man if it had not been unwrapped from the paper?" requires the model to reason the wrapping relationship between "the man", "the paper" and "the piston" as well as the causal connections in the challenging *counterfactual* setting. Additionally, we can observe from the case in the seventh row that our GROUNDMORE includes spatiotemporal grounding context as well as motion-related attributes understanding. The answer to the question "Who might not have fallen into the blue cushion on the wall if he had not tripped while trying to defend?" can only be determined at the end of the video clip. For the question "Who is the more offensive player?", the model must infer motion-based implicit attributes from the video sequence, demonstrating a strong need for world-level commonsense reasoning ability. These details further demonstrate the complex motion reasoning context of our GROUNDMORE.

Besides, the raw videos are processed into individual frames and stored in a folder named with the format "youtube_id_start-time_end-time". The annotation is in a JSON format, structured as follows:

```json
{
  "questions": {
    "1": {
      "action_end": "0:15",
      "action_start": "0:00",
      "answer": "The man",
      "obj_id": "1",
      "q_type": "Causal",
      "question": "Who uses the cut jug to scoop water out of the canoe?"
    },
    "2": {
      "action_end": "0:15",
      "action_start": "0:00",
      "answer": "The cut jug",
      "obj_id": "2",
      "q_type": "Causal",
      "question": "What does the man use to scoop water out of the canoe
          ?"
    }
  }
}
```

Each entry in the JSON file consists of a series of questions associated with the video. Each question contains the following fields:

- `action_start` and `action_end` specify the time segment in the video corresponding to the action.
- `answer` provides the correct response to the question.
- `obj_id` uniquely identifies the object involved in the question.
- `q_type` indicates the question type, such as "Causal".
- `question` is the text of the question related to the action in the video.

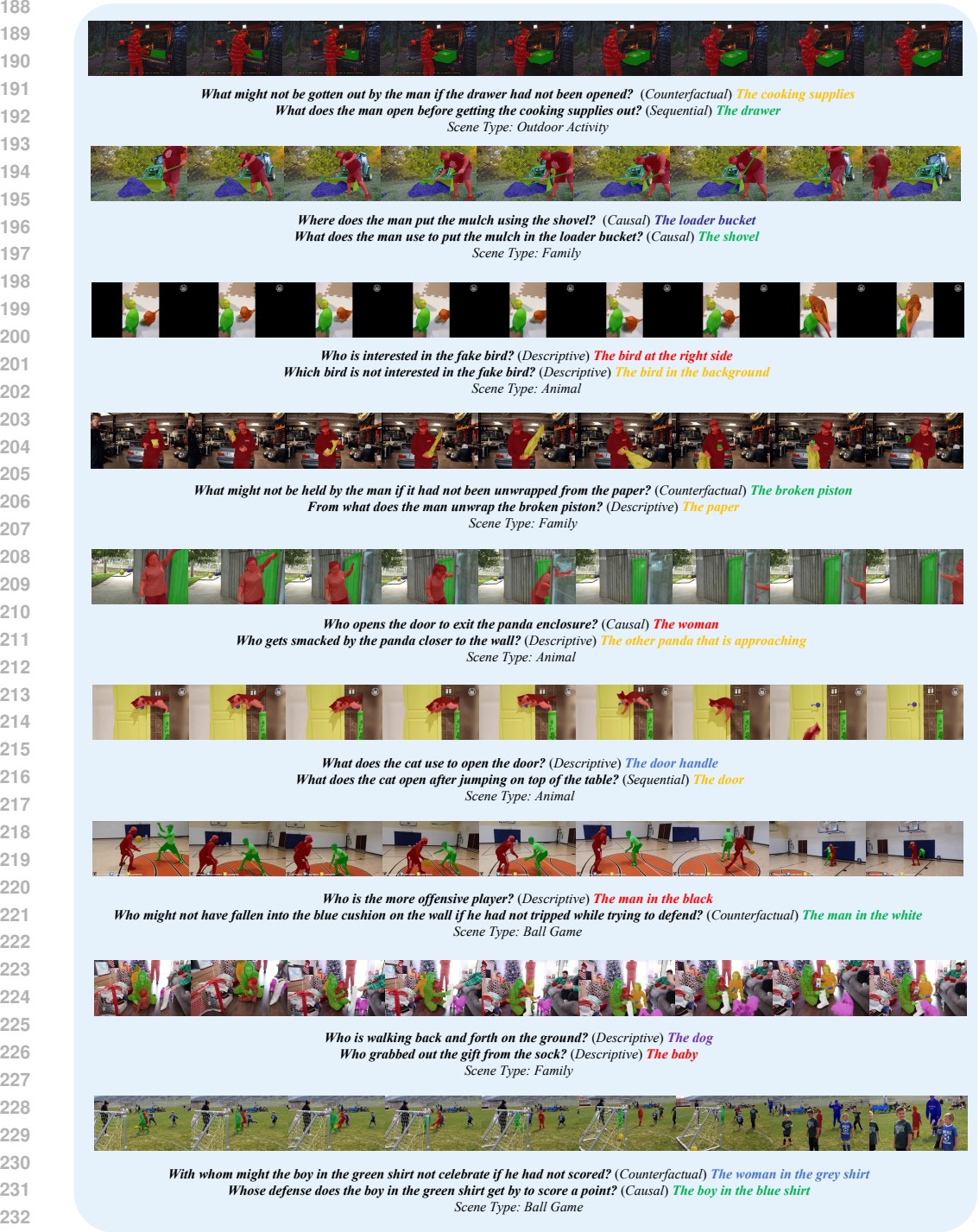

Figure 13: Additional Visualizations of our GROUNDMORE. We provide visualizations of videos alongside their corresponding segmentation masks, questions, answers (color corresponds to the segmentation masks), and scene types.

## A.4 DATASET NECESSITY

In previous MeViS (Ding et al., 2023), the more challenging motion expressions increase the difficulty of the dataset compared with previous benchmarks, since the target objects have to be distinguished from others by sophisticated motion understanding. In our GROUNDMORE, we not only consider the

Table 6: Comparison of explicit and implicit expression on MeViS valid-u.

| Expressions Type | J&F | J | F |
|---|---|---|---|
| original (explicit) | 40.23 | 36.51 | 43.90 |
| implicit | 32.33 | 28.81 | 35.86 |

abundant temporal reasoning clues in the motion expressions, we also take the *implicit reasoning* into account and we view it as a core challenge in Motion-Grounded Video Reasoning. Moreover, we hypothesize that containing motion expressions though, the object information in the input language in MeViS might still result in an identity leakage and make the model ignore the motion description but rely on the target information itself. To validate this, we made a modification on the original expressions in MeViS valid-u data so that the object name will be replaced by *"something"*, making the original explicit expressions into implicit ones. After this, we ran the evaluation process as usual and only found that the performance had an obvious drop, about 20% as shown in Table 6. In our GROUNDMORE, due to the fact that we intentionally omit the target identity by using the questions as our implicit expressions, we force the models to focus on the motion clues and perform reasoning before the segmentation process. In this way, the motion information is guaranteed to be leveraged. This interesting discovery in Table 6 not only demonstrates the weak implicit expression processing ability in existing models but also validates the necessity of our task and dataset, i.e., our implicit questions are not similar to the motion expressions.

A.5    IMPACT OF OBJECT NUMBERS

The number of objects will affect the results a lot, which is also consistent with the intuition that more objects in the videos will bring more difficulties in localizing target objects. Due to the time limit, we cannot obtain the overall analysis now, but we do obtain a subset results. Specifically, we randomly sample two subsets (containing 120 instances each) from GROUNDMORE, the first subset contains videos that include less than 3 objects, and the second one with more than 6 objects (we ignore visual-insignificant objects here). The results (MoRA zero-shot) are shown in Table 7.

Table 7: The impact of object numbers in GROUNDMORE.

| | J&F | J | F |
|---|---|---|---|
| #OBJ $\leq$ 3 | 15.56 | 16.93 | 14.18 |
| #OBJ $\geq$ 6 | 9.26 | 10.42 | 8.09 |

A.6    VIDEO LLMS IN TWO-STAGE BASELINES

Compared to the results in the main paper, we can still observe that SeViLA outperforms other video QA models in the two-stage setting. A key reason is that SeViLA generates concise and precise answers, avoiding the inclusion of redundant information that could negatively impact the performance of RefVOS models.

For example, given the question *"What does the man in white dribble?"*, the answers from the video QA models are as follows:

- **SeViLA**: "a basketball."
- **VideoChat2**: "The man in white is dribbling a basketball in the video."
- **VILA**: "The man in white dribbles the ball around the court while the man in black tries to block him."

Similarly, for the question *"Who snatches the ball after the man in grey accelerates towards him?"*, the answers are:

- **SeViLA**: "the man in red."

- **VideoChat2**: "The man in red snatches the ball after the man in grey accelerates towards him."
- **VILA**: "The man in grey snatches the ball after the man in red accelerates towards him."

## A.7    LIMITATION AND FUTURE WORK

Although our dataset has included a wide range of video scenarios, there are still many scenarios and motion types to be considered, e.g., motion in first-person-view videos. Besides, in the current version, we only consider single-object as target (even though multiple objects appear in the scene), which is less complicated than simultaneously grounding multiple targets.

Besides, we will also consider more modalities, such as audio (which could provide more nuance information beyond visual clues) and keypoint (which could introduce direct motion features), to construct more comprehensive training data as well as the evaluation benchmark.

## A.8    ETHICS STATEMENT

**Copyright and Fair Use Disclaimer.** The collection and use of GROUNDMORE are conducted in accordance with the principles of Fair Use [1] as outlined in U.S. copyright law, particularly for purposes such as research, scholarship, and commentary. The dataset is provided under a strict non-commercial use policy. Any use of GROUNDMORE must adhere to these restrictions, and users are prohibited from using the dataset in any way that may infringe on the rights of the original content creators. By accessing the dataset, users agree to comply with these terms and with the principles of Fair Use.

**Privacy Considerations.** Since GROUNDMORE includes segments from videos that may contain identifiable human faces and actions, we acknowledge the importance of addressing privacy concerns. The dataset is restricted to non-commercial use only, with the primary aim of advancing research and education. We have taken additional steps to ensure ethical standards are maintained by submitting the dataset for review by the Institutional Review Board (IRB) at our university, and the IRB submission is currently under review.

**License.** GROUNDMORE is distributed under the Creative Commons Attribution-NonCommercial 4.0 International License (CC BY-NC 4.0)[2]. This license allows others to remix, adapt, and build upon the dataset for non-commercial purposes, provided that appropriate credit is given. Commercial use of the dataset is strictly prohibited.

**Data Usage Responsibility.** We encourage all users of GROUNDMORE to adhere to ethical research standards, including fairness, transparency, and respect for individual privacy. Researchers are expected to consider the ethical implications of their work and to ensure that any models or technologies developed using GROUNDMORE do not inadvertently reinforce biases or infringe on individual rights.

---

[1]For more information on Fair Use, see https://www.copyright.gov/fair-use

[2]For more details on the license, see https://creativecommons.org/licenses/by-nc/4.0/

