# OpenReview forum: "Motion-Grounded Video Reasoning: Understanding and Perceiving Motion at Pixel Level"
_ICLR.cc/2025/Conference — ICLR 2025 Conference Withdrawn Submission_

### Official Review · Reviewer_86Cd · 2024-11-03

**Soundness:** 2
**Presentation:** 3
**Contribution:** 2
**Rating:** 3
**Confidence:** 5

**Summary:**

This paper introduces Motion-Grounded Video Reasoning, a novel task that requires generating spatiotemporal video segmentation masks as visual answers to motion-related questions, facilitating implicit reasoning about motion at the pixel level. To support this, the authors created the GROUNDMORE dataset, comprising 1,673 video clips with 243,000 object masks and questions in four categories: Causal, Sequential, Counterfactual, and Descriptive. They also present MORA (Motion-Grounded Video Reasoning Assistant), a baseline model combining multimodal reasoning, pixel-level perception, and temporal awareness to achieve state-of-the-art performance on GROUNDMORE, highlighting the increased complexity of this new task compared to existing benchmarks. This work aims to advance robust motion understanding in video reasoning segmentation for real-world applications.

**Strengths:**

The paper introduces a new, challenging task in motion-grounded video reasoning, which addresses the limitations of existing datasets by incorporating more complex motion-based interactions. This approach enables a more realistic scenario for evaluating video reasoning capabilities.

**Weaknesses:**

The paper would benefit from discussing some failure cases or limitations of the MORA model. Including challenging or negative examples in the experimental section could provide a more comprehensive evaluation and highlight areas for potential improvement.

While the paper proposes a new dataset and method, it could strengthen its contributions by comparing the model's performance not only on GROUNDMORE but also on existing, related datasets. This would help in better situating the improvements MORA brings within the broader research landscape.

I understand the segmentation mask of video frames is relevant to motion, but why the authors emphasise {Causal, Sequential, Counterfactual, and Descriptive} pair. Causal and Counterfactual seem irrelevant to segmentation mask.

**Questions:**

Could the authors provide additional details regarding the quality control measures for the GROUNDMORE dataset annotations? Specifically, information about the annotation guidelines, inter-annotator agreement, and methods used to ensure consistency would add clarity on dataset reliability.

Additionally, it would be helpful to discuss the specific benefits of MORA in comparison to other state-of-the-art models. A more detailed comparison could highlight MORA’s unique strengths in handling spatiotemporal reasoning and motion understanding relative to existing approaches.

---

### Official Review · Reviewer_iN3a · 2024-11-03

**Soundness:** 3
**Presentation:** 3
**Contribution:** 3
**Rating:** 6
**Confidence:** 4

**Summary:**

This paper introduces a new task called Motion-Grounded Video Reasoning, that requires understanding the concept of motion at the fine-grained pixel level in videos. In contrast to conventional video question answering tasks and benchmarks, this task goes beyond traditional spatiotemporal grounding by requiring models to generate visual answers, specifically video segmentation masks, in response to motion-related questions. To facilitate further research in this direction, the paper also introduces a new evaluation benchmark called GROUNDMORE, which includes 1,673 video clips and over 243,000 object masks across four types of questions: causal, sequential, counterfactual, and descriptive. This task requires a video understanding model to perform spatiotemporal grounding of queries and referred concepts.

Finally, to establish the difficulty of this task, the authors also presents a baseline model, named Motion-Grounded Video Reasoning
Assistant (MORA). The proposed MORA approach is a modular approach that combines the reasoning capabilities of a multimodal large language model (MLMM) with the segment anything model (SAM) for spatial grounding, along with a lightweight temporal localization head. MORA outperforms existing visual grounding models on GROUNDMORE, highlighting the dataset's challenge and its emphasis on complex motion reasoning and pixel-level temporal perception​. Experimental results demonstrate the effectiveness of the proposed MORA approach, where it outperforms state-of-the-art models by a significant margin.

**Strengths:**

1) In terms of significance, the paper addresses an important task that aims to extend the capability of existing video and language understanding models beyond answering questions with textual responses but also grounding them in pixels. By combining pixel-level video segmentation with complex spatiotemporal reasoning, the paper positions GROUNDMORE as a benchmark for developing more sophisticated multimodal models that can understand nuanced, temporally grounded motion. The introduction of a challenging new baseline model (MORA) further underscores the work's value, as it highlights significant room for advancement in video-based multimodal AI. This research could influence subsequent work in fields such as autonomous driving, surveillance, and assistive technologies, where understanding both spatial and temporal nuances of motion is critical. This is an important extension to current works such as LITA [1], which only focuses on temporal grounding.

[1] LITA: Language Instructed Temporal-Localization Assistant, ECCV 2024

2) Regarding the quality, the GROUNDMORE dataset appears to be a high-quality contribution due to its size and diversity. Furthermore, the questions have been curated into causal, sequential, counterfactual, and descriptive categories. This can be especially helpful for understanding how existing approaches per on different aspects of reasoning about motion.

3) In terms of clarity, the paper is well-written and motivated. The model figures are informative and especially helpful in helping the reader to understand the different stages of the data curation process as well as the intuition behind each component of the model.

**Weaknesses:**

1) While the proposed GROUNDMORE dataset is a nice contribution, it may be relatively limited in the variety of videos due to them being sourced only from the categories of family, animal, ball games, and outdoor activities. It could be helpful to expand the dataset to include scenes with multiple moving objects such as multiple cars. This would be helpful for evaluating how robust MLLMs are to different contexts of motion reasoning.

Moreover, it is mentioned in line 273 of the submission that the authors selected short clips which contain high degrees of motion semantics. However, it is not explicitly defined how the degree of motion semantics is actually being computed, such as leveraging optical flow information or other algorithms.

2) Given the video nature of this task, it is not well-motivated in the writing why the authors build their proposed MORA approach off the image-based reasoning segmentation framework LISA instead of the more recent video version, VISA [2]. In theory, the VISA model is trained with the objective of being able to perform spatiotemporal grounding of language queries in videos and should be able to generalize much better to the proposed task.

[2] VISA: Reasoning Video Object Segmentation via Large Language Model, ECCV 2024.

**Questions:**

NA. See weaknesses.

---

### Official Review · Reviewer_8aDx · 2024-11-06

**Soundness:** 3
**Presentation:** 3
**Contribution:** 3
**Rating:** 6
**Confidence:** 4

**Summary:**

This paper introduces a new task of motion-grounded video reasoning which requires answering a input text question with an output video segment. This tests both high level reasoning and low-level motion understanding and pixel level capabilities. The paper introduces a dataset as part of this contribution called GroundMore, and a baseline (MoRA) that achieves state of the art performance, along with a detailed analysis of existing benchmark methods on the task.

**Strengths:**

1) The paper is very thorough and introduces a new task, supporting dataset and model. It does everything that in my mind is required of a dataset/benchmark paper.
2) The analysis of existing baselines is extremely thorough and provides great context for what works and what doesn't on this task.
3) The appendices of the paper are very comprehensive and include all the details that one might need when asking about the dataset itself.
4) The presentation quality is strong and makes the paper easy to understand and read, and I really appreciate all the diagrams and details about the distribution of verbs, concepts, scenes and words in the dataset in the appendix.

**Weaknesses:**

1) I think the definition of "motion understanding" is very vague. The paper seems to equate "action" with "motion" which is not particularly useful - shouldn't motion understanding encode an understanding of *how* something moves, rather than just what something is doing? As an example, answer "Who fed the dog with a piece of dog food after taking the dog food out from the cabinet?" does not require any understanding of how the dog food was taken out, how the feeding was done, etc. Things like "Feeding a dog" or "taking out dog food" have been studied as part of temporal action localization or action recognition. In this sense, I'm not sure this dataset really tests or requires true motion understanding. I'd love some clarification on this point. Another examples is from Figure 7, the most common verb is "use". I don't really see how segmenting a person using something requires motion understanding.

2)  I'm not convinced the questions really require motion understanding. For example, from Figure 2, "Who points at the man in the orange shorts because the man in the orange shorts fouled?" doesn't require understanding that the man in orange shorts fouled. You just need to identify that someone is pointing at the man in orange shorts. Similarly, "Who drops the purple broom after seeing the panda would not leave the basket?" just requires finding a part where the woman drops a purple broom. Much of the paper states that the questions require understanding of motion semantics - this doesn't really seem clear to me, and I think needs more explanation about what that means. To me, it seems like the task is mostly about 1) identify an object with some reasoning module and then 2) performing video object segmentation. If this is the case, then all of the "motion understanding" component is through video object segmentation, and all the baselines proposed (along with MoRA) treat the two tasks totally independently, which runs contrary to the narrative from the introduction of the paper. At the end of the day, this then just becomes a segmentation benchmark and the "motion understanding" is essentially how strong the segmentation method is.

3) Using J&F doesn't seem like the right metric choice. this is essentially grading most of the output on the segmentation quality. Shouldn't there be some component that measures where the correct object was segmented? Is there a way to account for this in the benchmark? This seems to accoutn for the difference between each reasoning method (ViLA, SeVILa, VideoChat2) with the same segmentation method.

**Questions:**

1) Where were the videos and captions sourced from? it's not clear from the text.

2) The questions seem really overly wordy and not natural, eg "Who needs to be got rid of defense from by the man in black or he cannot shoot the basketball?". Why was this design choice made? Is every part of this sentence required? Why wouldn't more natural language make sense here?

3) Why weren't benchmarks with proprietary (gpt-4o, claude, etc) models that have vision capabilities benchmarked? A simple benchmark could be asking gpt-4o to identify the object of interest, then feeding that output directly to SAM2 or the other segmentation methods used. This might be a stronger baseline than MORA or SeViLA-based methods, since the propriety models have much stronger reasoning capabilities and can ingest multiple frames.

---

### Official Review · Reviewer_mqJ7 · 2024-11-08

**Soundness:** 2
**Presentation:** 3
**Contribution:** 2
**Rating:** 5
**Confidence:** 4

**Summary:**

This paper mainly introduces a new dataset called GroundMoRe for motion-centric pixel-level language grounding in videos. It collects a large number of video clips with question-answer pairs and object masks and also proposes a new baseline method MoRA to handle the new task on the new dataset. Extensive statistics and experiments have shown the advantages of the dataset and the effectiveness of the proposed method.

**Strengths:**

1.The paper is well-written and clearly presents the motivations of the proposed dataset as a new motion-centric video benchmark. And there are sufficient examples and figures given in the manuscript to illustrate the dataset characteristics.

2.I think the new task Motion-Grounded Video Reasoning proposed in this work is a more practical setting to benchmark the video multimodal models, since it considers the capability of temporal localization which is commonly neglected by previous methods.

3.The proposed baseline model is reasonable and shows effectiveness in addressing Motion-Grounded Video Reasoning on GroundMoRe.

**Weaknesses:**

1.Although one of the most significant contributions claimed by the authors is to introduce implicit reasoning into the video question-answering and segmentation models, this idea has been previously explored by several works like VISA [1] and ViLLa [2]. It seems that there is no obvious difference in terms of the introduction of implicit textual inputs compared to these existing works, and this makes the contribution and novelty of this work not that impressive.

2.In terms of the Sequential question type in the proposed GroundMoRe dataset, I have some concerns on the effectiveness of it for truly reflecting the model's ability to reason about the temporal relations of motions. For example, as shown in Figure 1, the query input "Who dribbled the ball before he accelerates passing the man in pink shorts?" involves two consecutive motions, but there is only one motion of "dribbling the ball" in the video clip so the model can easily localize the first motion without looking at the second motion to find the correct answer, while this behavior cannot be regarded as temporal relation reasoning. I think similar problem will also occur in other examples mentioned in the manuscript like "the woman opened the refrigerator before taking out the milk". In my view, some samples should be explicitly constructed for this question type, where a same motion occurs twice but one of them is accompanied with another context motion, so the model has to rely on the temporal relation with the context motion to localize the answer.

3.To my understanding, the most interesting contribution of this work is to take the temporal grounding into consideration during the benchmarking process. However, it seems that the evaluation and investigation of temporal grounding capabilities are quite simple. For example, I don't see the authors discuss any metrics regarding the temporal grounding capabilities, such as the typically adopted temporal Intersection over Union (tIoU). Actually I think there should be a more comprehensive metric considering the spatio-temporal localization ability in this work, which can be derived from a simple extension to the existing vIoU metric proposed in Spatio-Temporal Video Grounding [3, 4]. But the current metrics discussed in this work are still focusing on the spatial segmentation.

4.According to the experimental results presented by Table 4, I feel kind of confused that for the Descriptive question type, the performance improvement is still remarkable when replacing the implicit questions with the referring expressions, i.e., the first row and second row for all methods in Table 4, and sometimes this improvement is even more obvious than other question types. Given that the Descriptive questions already convey a lot of description details, why could this phenomenon happen? Intuitively I think the difference between a descriptive question and a referring expression is quite minor, maybe there is just something like whether the concrete subject word is mentioned or not comparing these two kinds of textual inputs?

References

[1] VISA: Reasoning Video Object Segmentation via Large Language Models, Yan et al.

[2] ViLLa: Video Reasoning Segmentation with Large Language Model, Zheng et al.

[3] Where Does It Exist: Spatio-Temporal Video Grounding for Multi-Form Sentences, Zhang et al.

[4] Human-Centric Spatio-Temporal Video Grounding With Visual Transformers, Tang et al.

**Questions:**

Please refer to the weaknesses.

---

### Note · Authors · 2024-11-14

I have read and agree with the venue's withdrawal policy on behalf of myself and my co-authors.